# Modifiable Host Factors for the Prevention and Treatment of COVID-19: Diet and Lifestyle/Diet and Lifestyle Factors in the Prevention of COVID-19

**DOI:** 10.3390/nu14091876

**Published:** 2022-04-29

**Authors:** Sawako Hibino, Kazutaka Hayashida

**Affiliations:** 1Y’s Science Clinic Hiroo, Medical Corporation Koshikai, Tokyo 106-0047, Japan; 2Department of Clinical Gene Therapy, Graduate School of Medicine, Osaka University, Osaka 565-0871, Japan; 3Division of Pulmonary Medicine, Boston Children’s Hospital, Boston, MA 02459, USA; kazutaka.hayashida@childrens.harvard.edu

**Keywords:** COVID-19, prevention, nutrients, functional food, lifestyle, dietary habits

## Abstract

Many studies have shown that the immune system requires adequate nutrition to work at an optimal level. Not only do optimized nutritional strategies support the immune system, but they also reduce chronic inflammation. Nutritional supplements that are recommended for patients with critical illnesses are thought to also be effective for the coronavirus disease 2019 (COVID-19) patients in the intensive care unit. Some studies have recommended fresh fruits and vegetables, soy, nuts, and antioxidants, such as omega-3 fatty acids, to improve immune system activity. Although nutritional status is considered to be an important prognostic factor for patients with COVID-19, there is to date no sufficient evidence that optimal nutritional therapies can be beneficial for these patients. Some have argued that the COVID-19 pandemic is a good opportunity to test the effectiveness of nutritional intervention for infectious diseases. Many researchers have suggested that testing the proposed nutritional approaches for infectious diseases in the context of a pandemic would be highly informative. The authors of other review papers concluded that it is important to have a diet based on fresh foods, such as fruits, vegetables, whole grains, low-fat dairy products, and healthy fats (i.e., olive oil and fish oil), and to limit the intake of sugary drinks as well as high-calorie and high-salt foods. In this review, we discuss the clinical significance of functional food ingredients as complementary therapies potentially beneficial for the prevention or treatment of COVID-19. We believe that our review will be helpful to plan and deploy future studies to conclude these potentials against COVID-19, but also to new infectious diseases that may arise in the future.

## 1. Introduction

In December 2019, severe acute respiratory syndrome coronavirus 2 (SARS-CoV-2) was reported to have migrated from animals in the Huanan Seafood Market in Wuhan, Hubei, China, to humans, and then spread rapidly around the world [1]. Despite subsequent global efforts, the rapid spread of coronavirus disease 2019 (COVID-19) has significantly affected the world. In March 2020, the World Health Organization (WHO) declared a global health emergency along with the declaration of a pandemic [2,3], and by the end of 2020, several vaccines were established for use in different parts of the world, with more than 40 candidate vaccines tested in human trials and more than 150 tested in preclinical trials. The WHO has an updated list of vaccine candidates that are currently under evaluation [2].

As of December 2021, (SARS-CoV-2), the pathogen responsible for COVID-19 and the ongoing global pandemic, has claimed approximately 5.4 million lives and infected 276 million people [4]. Even after recovery, about 50% of patients experience sequelae, such as prolonged symptoms, with long-term medical, psychological, and economic consequences.

The causative virus, SARS-CoV-2, infects cells of the respiratory system mainly by droplet infection, causing respiratory diseases ranging from inflammatory symptoms in the upper respiratory tract to fatal severe pneumonia. It is well known that the poor nutrition status of a host, including having diabetes, hypertension, and obesity, is a potential risk factor for severe respiratory diseases and comorbidities, all of which increase the risk of severe illness, hospitalization, and death in COVID-19 patients [5,6].

At present, measures against COVID-19, such as wearing masks, sanitizing one’s hands, and avoiding the Three Cs (closed spaces, crowded places, and close-contact settings) are being practiced in Japan as preventive methods to reduce the chance of exposure to the causative virus. Vaccination is also being deployed and has shown significant effectiveness in preventing infection as well as severe illness and mortality. However, the emergence of immune evading variants and the waning of immunity over the long term has been considered to erode the persistent protection from the vaccines. Therefore, it is still important to take measures to increase resistance to viral infections in addition to vaccination. Adequate nutrition in the host is essential for maintaining the body’s defense against viral infections, as under-nutrition or nutritional disorders can lead to decreased immunity and an increased risk of contracting infectious diseases [7,8]. In particular, an adequate diet and intake of vitamins, minerals, and other functional food ingredients can be another important modality to fight against COVID-19. To date, there is insufficient evidence to support the recommendation of dietary modification for or against COVID-19. However, some preliminary clinical studies have already shown the benefits of vitamins, minerals, fatty acids, probiotics, and herbs against COVID-19. In addition, a study of 440,000 people in the UK (*n* = 372,720), the USA (*n* = 45,757), and Sweden (*n* = 27,373) found that users of probiotics, omega-3 fatty acids, multivitamins, and vitamin D supplements had a significantly lower risk of SARS-CoV-2 infection than non-users [9].

The pandemic has abruptly caused fundamental changes in habits and lifestyles, with social distancing and quarantine restrictions having a major impact on the lives of people in terms of their diet and daily activities. Numerous reports have assessed changes in diet [10,11,12], such as those focusing on children and adolescents in China [13], adolescents and medical students in Croatia [14], adults in Israel [15], adults in Brazil [10], a single country in Europe [16,17,18], and children or adults with comorbidities [19,20,21]. One study reported a comparison of dietary changes among adolescents in various Ibero-American countries during the pandemic [22]. Therefore, the impact of these changes related to the pandemic on human wellbeing should also be studied.

## 2. Importance of a Healthy Lifestyle and Dietary Patterns

A recent editorial in the journal *Nutrients* emphasizes how effective non-pharmacological interventions combined with promoting a healthy lifestyle and dietary patterns can improve overall health and reduce the risk of infection from SARS-CoV-2. This also highlights the potential reduction in the risk of infection and adverse effects of COVID-19 by a healthy lifestyle and diet. COVID-19 is still affecting our daily lives in 2022 and will continue to do so for the foreseeable future. Scientific advances have focused on the development of vaccines, the production of therapeutic agents, and the promotion of non-pharmacological interventions to reduce disease burden. However, we are currently experiencing a recurrent wave of pandemics across different parts of the world. Therefore, the active promotion of healthy lifestyle patterns can be an alternative approach to prevent the diverse sequelae of COVID-19, as an underrated mitigation strategy, along with non-pharmacological interventions. This will be important regardless of vaccination status, as obviously lifestyle-related factors such as obesity, hypertension, and diabetes are the risk factor for severe COVID-19 illness [23,24]. Furthermore, lifestyle and dietary modifications can improve the overall health status, which also prevents diseases other than COVID-19 associated ones.

## 3. Importance of Diet for the Prevention of Viral Diseases

Although there is no direct evidence to support the importance of diet for preventing COVID-19 infections and severe disease, and as a countermeasure against sequelae, the usefulness of a diet consisting mainly of plant-based foods and Mediterranean foods can be inferred [25]. Mediterranean foods contain functional food components with antioxidant and anti-inflammatory properties, which may aid in reducing the symptoms of COVID-19 sequelae. Moreover, diet modification has been linked to the incidence of lifestyle diseases such as obesity, hypertension, and diabetes, which are the risk factor of severe COVID-19 [23,24]. Diet can affect COVID-19 directly or indirectly.

## 4. The Mediterranean Diet

The Mediterranean diet is rich in functional components derived from vegetables, fruits, seeds, fish, and olive oil, and has been suggested to reduce COVID-19 symptoms via anti-inflammatory and antioxidant effects [26]. Several studies have confirmed an inverse association between adherence to a Mediterranean diet (MD) and overall cancer-associated mortality. A healthy MD is an appropriate combination of high-quality foods based on their macronutrient and micronutrient content and is free of contaminants. According to recent information, MD can be an important factor against immune and inflammatory responses, such as those that occur in cancer. In particular, the MD has been associated with reduced mortality from obesity, type 2 diabetes, mild inflammation, cancer, Alzheimer’s disease, depression, and Crohn’s disease [27,28]. Although Mediterranean countries such as Italy and Greece were severely affected by COVID-19 in the early days of the pandemic, the effects of the MD diet on COVID-19, particularly on long-term sequelae, should be investigated and determined.

Vegetarian diet: A nutritious vegetarian diet also aids in reducing the risk of lifestyle-associated and chronic diseases [25,29]. Compared with animal-based foods, consuming plant-based foods changes the gut microbiota in an ideal direction, suggesting a risk-reducing effect of COVID-19 via intestinal immunity [25,30].

Anti-inflammatory diet: An anti-inflammatory diet is a diet that includes foods common to Mediterranean and vegetarian diets as well as omega-3 fatty acids and is considered as an integrative medicine to combat COVID-19. In addition to foods common to the Mediterranean and vegetarian diets, an anti-inflammatory diet also specifies the use of fish rich in omega-3 fatty acids, mushrooms, green tea, dark chocolate, and supplements [31]. The effect of these specific ingredients must be determined in well-designed clinical studies in the future.

## 5. Low COVID-19 Infection and Mortality in Rice-Eating Countries

It is interesting to note that in general, the rates of infection and mortality from COVID-19 are much lower in East Asian countries than in other parts of the world. A study by Watanabe et al. has shown that countries in which rice is the staple food have lower COVID-19 infection rates than countries in which wheat is the staple food [32]. In this study, Dr. Watanabe and his colleagues conducted a statistical analysis of the correlation between rice consumption and the number of COVID-19 cases (as of 6 June 2021) in 19 countries on five continents (Canada, the United States, Mexico, Argentina, the United Kingdom, France, Italy, Germany, Spain, Russia, China, Japan, India, Indonesia, South Korea, South Africa, Turkey, Saudi Arabia, and Australia). They found a negative correlation between rice consumption per capita (kg) and the number of COVID-19 cases per million population (coefficient of determination: 0.5916). On the other hand, a positive correlation between wheat consumption and the number of COVID-19 cases was found (coefficient of determination: 0.4879). This suggests that a diet based on brown rice, rice bran, and whole grains may act as a countermeasure to COVID-19. However, other confounding factors such as overall lifestyle, other food sources, or people’s behavior must be involved to make this difference. Perhaps it is too much to conclude that eating rice directly suppresses COVID-19, although it is a very interesting hypothesis. The concept that dietary culture can determine susceptibility against SARS-CoV-2 is quite attractive and further studies, such as interventional cohort studies, will clarify the direct relationship between rice-eating and lower risk of COVID-19 infection. 

## 6. Functional Food Components

Clinical studies are currently being carried out to evaluate the effectiveness of components in functional foods against the long-term effects of COVID-19.

Coenzyme Q_10_: Mitochondria are the energy factories in cells, and they act as control towers to ensure that the immune system works correctly, affecting the activation of all cell types, including B cells that produce neutralizing antibodies. Therefore, all major natural immune pathways are dependent on mitochondria. However, as the number of mitochondria tends to decrease with age, supplementing the body with coenzyme Q_10_ (CoQ10), which assists mitochondrial function, is considered to be important [33]. The intake of CoQ10 was also shown to increase the levels of Immunoglobulin A (IgA), which is part of the natural immune system and is abundant in the oral cavity and was found to promptly attack invading SARS-CoV-2 in the oral cavity and prevent it from adhering to mucous membranes and entering the body [34].Therefore, the consumption of commercially available CoQ_10_ in the form of supplements or functional foods (particularly in its reduced state), such as the prescription drug ubidecarenone (product names include Neuquinon), may be effective. However, there is no evidence that Coenzyme Q supplementation is beneficial against COVID-19. We should wait for the recommendation to use this until a clinical study showing the effect comes out. A randomized placebo-controlled double-blind clinical trial testing the effect of Coenzyme Q on long-term COVID-19 symptoms is underway [35] (as of March/2022). The result will be interesting.

## 7. Importance of Vitamins and Minerals

Recently, the European Society for Clinical Nutrition and Metabolism proposed a list of 10 practical recommendations for caring for COVID-19 patients [36]. The recommendations include measures for preventing malnutrition by providing key nutrients in appropriate quantities to keep energy, protein, fat, and carbohydrates at the required levels. To prevent viral infection, it is also important to ingest sufficient vitamins, minerals, etc.

Low levels of micronutrients, such as vitamins A, E, B_6_, and B_12_, zinc, and selenium have been reported to be associated with unfavorable clinical outcomes during viral infection [37]. A recent review by Zhang and Liu [38] demonstrated that not only vitamins A and D, but also vitamins B and C, omega-3 polyunsaturated fatty acids, and trace elements (selenium, zinc, and iron) must be considered when evaluating trace nutrients in COVID-19 patients. In a recent small-scale study of the nutritional state of COVID-19 patients in Korea, significant vitamin D and selenium deficiencies were found, regardless of the presence or absence of pneumonia [6]. In this study, serum levels of vitamins B_1_, B_6_, B_12_, D, folic acid, selenium, and zinc were measured in 50 COVID-19 patients, and 76% of the patients were found to be deficient in vitamin D, while 42% were deficient in selenium [6]. Selenium deficiency has been associated with high COVID-19 mortality rates [39], and sufficient selenium levels are thought to be important for adequate recovery from COVID-19 [40].

## 8. Importance of Vitamin C against COVID-19

Vitamin C treatment has antiviral effects, and clinical trials have shown that high-dose vitamin C is effective against the common cold [41,42]. Patients who received high-dose vitamin C treatment (1000 mg of vitamin C once an hour for the first six hours, then three times a day for the next three days) showed reduced flu and cold symptoms compared with the control group [41]. A meta-analysis showed that high doses of vitamin C given at the onset of a common cold shortened disease duration and relieved symptoms such as chest pain, fever, and the chills [42].

There is limited evidence to support the use of vitamin C against COVID-19. For example, a case using high-dose vitamin C treatment against COVID-19 has been reported. A 74-year-old woman with COVID-19 developed acute respiratory distress syndrome and septic shock. She was treated with high-dose intravenous vitamin C (11 g/day for 10 days) and showed rapid recovery [43]. In another study, 17 COVID-19 patients were given intravenous vitamin C at a dose of 1 g every eight hours for three days. After the vitamin C treatment, levels of inflammatory markers, such as ferritin and D-dimer decreased, and some of their previous inhalation oxygen levels (the requirement for oxygen inhalation decreased) were reduced [44]. These studies suggest that the administration of vitamin C may increase the survival rate of COVID-19 patients by suppressing the overactivation of the immune response [45]. In contrast, an observational study of more than 45,000 residents from the UK, US, and Sweden did not show any association between the use of vitamin C supplements and COVID-19 infection risk [9]. Moreover, daily supplementation of 8000 mg vitamin C on non-hospitalized COVID-19 patients did not shorten the symptom duration with regardless of combination to 50 mg of Zinc [46].

There are also studies examining the effect of vitamin C intravenous administration because a much higher concentration in plasma is expected. Intravenous administration of 10–20 g vitamin C daily improved oxygenation in 50 patients with moderate to severe COVID-19 [47]. In contrast, 24 g of intravenous vitamin C treatment on ICU patients did not improve the 28-day mortality rate [48]. Another study using 6 g intravenous vitamin C daily on severe COVID-19 patients did not affect the mortality and duration of ICU stay [49].

Currently, available evidence cannot recommend using oral or intravenous vitamin C on COVID-19. Further large-scale clinical studies should be conducted.

## 9. Importance of Vitamin D against Viral Diseases

Appropriate vitamin D levels in the body can be attained by consuming sufficient vitamin D and being exposed to sunlight. The risk factors for vitamin D deficiency are associated with older age, smoking, obesity, and chronic conditions, such as diabetes and high blood pressure [50].

As reported in the National Health and Nutrition Survey 2001–2006, 25-hydroxy vitamin D levels were inversely correlated with the incidence of acute respiratory infections [51]. In other words, sufficient concentrations of 25-hydroxy vitamin D correlated with a lower risk of acute respiratory infection in adults [52]. Furthermore, sufficient blood serum levels of 25-hydroxy vitamin D were inversely correlated with the risk of viral respiratory tract infections in children [53].

The association between vitamin D levels and COVID-19 has been investigated in studies [54,55]. We have mixed results. A cross-sectional study on COVID-19 patients in Wuhan matched with healthy control identified the Vitamin D deficiency (<30 nM vitamin D) independently associated with COVID-19 severity [56]. Another prospective study including severe COVID-19 patients found that hypocalcemia and hypovitaminosis D were common among the severe COVID-19 and associated with a poor prognosis [57]. Another study demonstrated that Serum 25(OH) vitamin D was independently associated with mortality in COVID-19 patients [58]. In a meta-analysis of 11 studies on 360,972 COVID-19 patients, 37.7% of the patients were vitamin D deficient, and 32.2% had insufficient levels of vitamin D. Furthermore, the risk of COVID-19 was significantly higher for patients with low vitamin D levels [59]. A separate meta-analysis of 1368 COVID-19 patients suggested that low vitamin D levels significantly correlated with poor prognoses [60]. The mortality rate of patients hospitalized with COVID-19 who have sufficient levels of vitamin D (serum 25-hydroxy vitamin D ≥ 30 ng/mL) was 5%. However, the mortality rate after 10 days of hospitalization for patients with severe vitamin D deficiency (serum 25-hydroxy Vitamin D < 10 ng/mL) was 50% [61]. There was a positive correlation between the rate of hospitalizations and ICU hospitalizations of patients with COVID-19 and vitamin D deficiency [62]. Furthermore, pre-infection vitamin D levels were found to be inversely correlated with the severity of COVID-19 symptoms [63].

On the other side, a large-scale Genome-wide association study on 443,734 participants could not find the association between genetic variants to determine the 25 OH vitamin D serum level and COVID-19 susceptibility, or severity [64]. Another study on the UK biobank of 502,624 participants found that vitamin D concentration is not associated with COVID-19 infection [65]. It is still controversial whether the serum level of vitamin D is the prognostic factor for COVID-19 infection or severity.

As vitamin D can be synthesized when skin is exposed to sunlight, living at high latitudes is a risk factor for vitamin D deficiency [66]. The mortality rate, which correlates with the prevalence rate of COVID-19, was significantly higher at high latitudes (latitude ≥ 37°) than at low latitudes (latitude < 37°) [67]. The average annual exposure to sunlight was negatively correlated with the rate of COVID-19 mortality [68]. However, the daily supplementation of vitamin D in countries at higher latitudes is very common. Therefore, it is still not clear that vitamin D deficiency is responsible for the correlation.

Even though we have mixed results to indicate the association between the vitamin D levels and COVID-19, it is reasonable to consider that vitamin D is an adjuvant treatment option for COVID-19 patients. Active vitamin D increases the expression of vitamin D-regulated genes in human tracheobronchial epithelial cells, such as cathelicidin and the toll-like receptor co-receptor cluster of differentiation (CD)14 [69]. Double-stranded RNA produced by most viruses can increase the expression of 1α-hydroxylase, leading to increased production of active vitamin D and the expression of cathelicidin in human tracheobronchial epithelial cells [69]. Although SARS-CoV2 is a single-strand RNA virus, it is still elusive how the innate immunity such as cathelicidin is involved in the protection against COVID-19. Adequate vitamin D might prevent the invasion of coronavirus by enhancing physical barriers and increasing the production of antimicrobial peptides in the lung epithelium [70]. Furthermore, vitamin D may prevent cytokine storms in COVID-19 patients. COVID-19 can cause a cytokine storm and immunogenic damage to the endothelium and alveolar membrane, and this could contribute to mortality [71]. Compared to patients with moderate symptoms, patients with severe COVID-19 symptoms express high levels of inflammatory cytokines, such as IL-6. Furthermore, the increased levels of IL-6 among COVID-19 patients with severe symptoms correlate with the detection of SARS-CoV-2 nucleic acid in serum [72]. Vitamin D may reduce the production of inflammatory cytokines, such as TNF-α, IL-6, IL-1β [73], IL-12, and **i**nterferon-**γ**(IFN-γ) [74]. The anti-inflammatory effect of vitamin D may be due to the inhibition of nuclear factor κB (NF-κB) activation [75]. Vitamin D receptors interact with κB (and IκB) kinase β inhibitors to inhibit NF-κB activation, and the interaction is known to be augmented by vitamin D [76].

Some studies are showing the effect of vitamin D supplementation on COVID-19. UK residents having vitamin D supplements more than three times a week for at least three months had a 9% reduction in COVID-19 infection risk [9]. Another observational study from the UK on in-patients with COVID19 showed that high dose vitamin D3 booster therapy reduced mortality by 3–7 fold regardless of serum vitamin D levels [77]. Furthermore, a cohort study from Spain on more than 4.5 million residents found that vitamin D3 supplementation achieving 25OHD levels ≥ 30 ng/mL was associated with a modest reduction of infection, severe illness, and mortality. Interestingly, there were no reductions with vitamin D2 supplementation unless they compared to vitamin D deficient patients [78]. However, one RCT designed to see the effect of a single oral high dose of vitamin D3 (200,000 IU) on hospitalized patients with moderate to severe COVID-19 did not find any reduction of hospital length of stay by vitamin D supplementation compared to placebo [79].

Such a mixed situation of conflicting results prompts additional large-scale clinical studies to clarify the effects of vitamin D on COVID-19 [65]. To date, concrete evidence that supplemental vitamin D prevents COVID-19 infection and ameliorates disease symptoms has yet to be established. Therefore, The National Institute for Health and Care Excellence (NICE) does not recommend using vitamin D to treat or prevent COVID-19.

## 10. Importance of Selenium in the Immune System in the Prevention of Viral Infections

Selenium deficiency exacerbates the virulence and progression of viral infections, such as influenza A [75,80] and coxsackievirus B3 [81]. Selenium-deficient mice developed more severe lung lesions due to influenza virus infection than selenium non-deficient mice [75]. Viruses collected from the lungs of selenium-deficient mice on day 5 after the infection had mutations in their genome and were more virulent [75]. Selenium is important for the function of cytotoxic effector cells, such as CD8^+^ T cells and natural killer cells. TNF-α and IFN-γ have antiviral effects against the influenza virus in CD8^+^ T cells [82]. Selenium supplementation increased the plasma levels of TNF-α and IFN-γ in mice infected with the influenza virus [83]. These findings suggest that selenium deficiency may be a risk factor for COVID-19 infection or mortality. Indeed, a cross-sectional study conducted in Germany showed that serum levels of selenium were significantly higher in COVID-19 patients who survived than in those that did not [37]. Another study also found that the rate of recovery from COVID-19 in Chinese patients was significantly associated with selenium levels [71]. Low plasma selenium levels were correlated with increased tissue damage, the presence of infection, and organ failure, as well as increased mortality in COVID-19 patients in the ICU. Additionally, plasma selenium levels were positively correlated with minimum platelet count, minimum plasma antithrombin activity, and protein C activity in COVID-19 patients [84]. However, blood selenium levels can be confounded by serum albumin levels, which are often lower in severe illness, because 37% of selenium is bound to albumin in the blood [85].

Blood clotting is one of the factors responsible for COVID-19 mortality [82,86]. Venous thromboembolism includes deep vein thrombosis, and pulmonary embolism is responsible for severe illness and mortality of COVID-19 patients [87]. It has been reported that venous thromboembolism occurred in 27% of COVID-19 patients in the ICU [88]. In another report, the cumulative incidence of venous thromboembolism in COVID-19 patients at 7, 14, and 21 days after ICU admission was 26%, 47%, and 59%, respectively [89]. The incidence was significantly higher in ICU patients than in patients in the general ward. Selenium deficiency can cause blood clotting [90] by increasing the ratio of thromboxane A2 to prostaglandin I2 [91]. This may suggest that selenium supplementation prevents venous thromboembolism in severe COVID-19 patients.

To date, there is no published study about selenium supplementation on COVID-19, although some clinical studies are ongoing.

## 11. Importance of Zinc in the Prevention of Viral Infections

Zinc is an important trace mineral, is involved in many biological processes including immunity, and is essential in both innate and acquired immune responses to viral infections [92,93]. The lack or deficiency of zinc has been implicated in smell and taste disorders associated with COVID-19. In a study conducted in Japan, a total of 251 asymptomatic, mildly ill, and moderately ill COVID-19 patients in hospitals or hotels were analyzed for subjective symptoms of smell and taste disorders, and 119 patients underwent smell and taste tests. The results showed that 37% of the patients had taste and smell disorders, 20% only had smell disorders, and 4% only had taste disorders. In this study, the rate of symptom improvement after one month was 60% for smell disorders and 84% for taste disorders, which is consistent with foreign reports. Therefore, the symptoms of taste and smell disorders are expected to disappear rapidly in most cases, as patients recover from COVID-19. Zinc is thought to be an important mineral in COVID-19 infection because of its immunomodulatory and antiviral properties. Zinc has been shown to inhibit the synthesis, replication, and transcription complexes of coronaviruses [94]. It is also known to directly interfere with viral replication and protein synthesis and shows beneficial and therapeutic effects on viral infections [92]. The immunomodulatory and antiviral properties of zinc may contribute to supportive therapy for COVID-19 patients. Therefore, several studies testing the efficacy of zinc supplementation as a treatment option against COVID-19 have been reported, including RCT studies. One RCT study with a relatively small sample size performed in Egypt showed that there is no significant benefit of adding zinc to hydroxychloroquine treatment with regard to the clinical outcomes of COVID-19 patients [95]. Another RCT study compared the effects of zinc or Vitamin C supplementation in the outpatient settings, in which the duration until symptoms started to improve was analyzed. There was no significant difference in the effects of zinc and vitamin C, although zinc caused nonserious adverse effects more frequently. Although there was no placebo control, this study suggests that zinc does not have stronger positive effects than vitamin C on symptom resolution [46]. An observation study also showed no significant reduction in in-hospital mortality by the administration of zinc [96]. In contrast, one retrospective cohort study, which has not been peer-reviewed, showed that zinc plus hydroxychloroquine treatment improved in-hospital mortality, although the patients were not randomly assigned to the treatments [97]. A case series of four COVID-19 patients treated with high-dose zinc also showed improvement in both clinical symptoms [98]. At present, the overall available evidence does not support the use of zinc as a treatment option against COVID-19. Furthermore, the COVID-19 treatment guidelines panel at the National Institutes of Health (NIH) recommends against using zinc supplementation above the recommended dietary allowance for the prevention of COVID-19, except in a clinical trial **(BIII)**. As the significance of zinc supplementation in the prevention of COVID-19 infection remains elusive, clinical studies to clarify the preventive role of zinc against COVID-19 should be conducted in the future.

## 12. Importance of Omega-3 Fatty Acids in the Prevention of Viral Infections

Monounsaturated fatty acids are contained in fish, nuts, and olive oil, and their low cholesterol content indicates a high antioxidant effect in fruits and vegetables. On the other hand, omega-3 fatty acids are polyunsaturated fatty acids, including eicosapentaenoic acid and docosahexaenoic acid, which are known to have an inhibitory effect on inflammation [99].

Interestingly, omega-3 fatty acids have been shown to demonstrate antiviral effects by inhibiting influenza virus replication [100]. According to a statement by the European Society for Parenteral and Enteral Nutrition, the administration of omega-3 fatty acids may improve oxygen supply in COVID-19 patients but has not been confirmed to date [36]. However, another study suggests caution regarding the use of omega-3 fatty acids in COVID-19 patients, as its abundance may increase cell membrane damage, which may then increase oxidative stress and inflammation [101].

Although it seems effective against COVID-19, EPA and DHA levels in blood cells did not associate with mortality risk from COVID-19 among 100 hospitalized patients [102]. However, a small study showed that severe COVID-19 patients who had omega-3 1 g for two weeks had a higher chance of survival at one month [103].

Omega-3 fatty acids may be beneficial in COVID-19 patients, but we need to confirm it in further clinical studies.

## 13. Other Nutritional Supplements and Their Effects on COVID-19

Licorice extract and hesperidin are two natural substances believed to prevent virus invasion through angiotensin-converting enzyme 2 and transmembrane protease serine subtype 2, which are two pivotal cellular proteins used by SARS-CoV-2 to enter mammalian cells. In other words, Licorice extract and Hesperidin can prevent the symptoms of COVID-19 from becoming worse. The mechanism may prevent SARS-CoV-2 from cellular invasion regardless of S protein mutation. Therefore, Licorice extract or Hesperidin might be useful as a potential preventive treatment against COVID-19. Indeed, glycyrrhizin (GA), the main component of licorice extract, was found to inhibit SARS-CoV-2 replication in a laboratory setting through inhibiting the viral proteinase 3C-like proteinase, which is also the target of the effective anti-viral pill Nirmatrelvir, which was developed by Pfizer [104]. GA also inhibits 11beta-hydroxysteroid dehydrogenase to increase the level of endogenous steroid hormones in vivo, which may limit excess inflammation, which causes severe illness, although its excess intake can cause hypertension by an increase in mineralocorticoids [105]. Furthermore, GA has been shown to be a direct inhibitor of high mobility group box 1 (HMGB1), which is a damage-associated molecular pattern responsible for inflammation and further tissue damage [106]. Even in severe cases of COVID-19, the attenuation of HMGB1 via GA may help prevent further tissue damage. However, evidence in clinical settings is lacking. Therefore, we should wait before using GA as a treatment against COVID-19 until clinical evidence is obtained because the long-term intake of GA can cause serious side effects such as pesudohyperaldosteronism, which is characterized by hypertension and hypokalemia [107].

The regular intake of highly nutritious foods leads to improvements in health. Some studies have suggested that consistently eating whole foods and plant-based foods improves the gut microbiota and supplies some of the essential phytochemicals required for the optimization of the function of minerals, vitamins, and other metabolic pathways. This may help the immune system to defend the body from pathogens [108,109].

Fruits and vegetables are rich in dietary fibers that support healthy gut microbiota, as well as antioxidant phytochemicals, such as alkaloids and flavonoids. Some vegetables, such as garlic, onions, ginger, turmeric, and berries are abundant in molecules that have anti-inflammatory, antioxidant, antiviral, and cell-protective effects [110].

The WHO has recommended that people eat beans, vegetables, and fresh fruits at home during quarantine [111]. Foods and appropriate nutritional supplements that can reduce SARS-CoV-2 infections and subsequent inflammation, oxidation, and cytokine storms should be investigated [112]. Further information regarding the association between COVID-19 and dietary supplements can be obtained from the NIH website.

## 14. Association between Viral Infections and Sleep

Interestingly, various studies have reported an association between sleep disorders and obesity, owing to the increased secretion of inflammatory cytokines via the increased visceral fat, which may contribute to changes in sleep-wake rhythms [113,114]. Furthermore, diet also appears to affect sleep quality. A cross-sectional study of 172 middle-aged adults reported that good sleepers had higher adherence to the Mediterranean diet and lower body weight, along with a lower body mass index (BMI) than sleep-deprived individuals [115]. Whereas consistent sleep strengthens the immune system [116], sleep deprivation is associated with the risk of chronic diseases [117]. One supplement for sleep disorders is valerian (Valeriana officinalis), which was found in a meta-analysis to improve sleep quality and duration [118]. A stronger effect of valerian was observed for subjects with a higher degree of insomnia. The high prevalence of sleep problems among the population that was found in a recent review can be explained by the COVID-19 pandemic and sleep-associated factors (e.g., delayed bedtimes, lights-out times, changes in sleep-onset times, and changes in wake-up habits owing to lockdowns and quarantines) [119]. One study showed that melatonin usage was associated with a 52% reduction of COVID-19 infection risk in African Americans [120], although it is unclear whether this reduction is caused by the improvement of sleep quality or not.

## 15. Association between Diet and Mental Health

A recent study showed an increase in unhealthy dietary behavior and a high frequency of depression and anxiety symptoms in Spain and Greece during the COVID-19 outbreak [121]. A higher prevalence of anxiety resulting from confinement during the pandemic was also observed in Saudi Arabia (27.7%) [122] and China (28.8%) [123]. In Greece, greater anxiety was associated with the increased consumption of meat products, as well as sweets in women [124]. A study in Iran found an inverse correlation between anxiety and the increased consumption of vegetables, fruits, and dairy products [125]. The fact that a higher frequency of vegetable and fruit consumption is positively correlated with subjective well-being is consistent with previous studies [125,126,127,128]. The COVID-19 pandemic has resulted in an urgent need to disseminate this health information to the public via the Internet and health collaborators [129,130]. For example, Stranges et al. [131] reported that people with reduced fruit and vegetable intake were more likely to have poorer mental health. In addition, a systematic review of 61 studies showed that a higher total intake of fruits and vegetables may promote higher levels of optimism and reduce psychological distress, thereby positively affecting mental health [128]. However, more research is needed to understand the correlation between mental health and the amount of fruit and vegetable intake. Recent findings on the association between unfavorable eating habits, such as skipping breakfast, and well-being add to the existing evidence that skipping breakfast is associated with worsened physical and mental health outcomes [127,132,133]. For example, breakfast skippers were found to have a significantly lower physical and mental health-associated quality of life than breakfast eaters in a population-based study in Taiwan [134]. Furthermore, a large cohort study showed that a dietary pattern characterized by skipping or delaying breakfast was associated with mood disorders in Australian adults [135]. However, further research is needed to clarify the association between mental health and breakfast eating patterns.

## 16. Association of COVID-19 with Obesity Due to Smoking and Lack of Exercise

There is a significant association between SARS-CoV-2 infection and air pollution, including smoking, with more severe COVID-19 symptoms occurring in smokers [136]. It has also been suggested that physical inactivity affects both body fat and appetite dysregulation [137]. Maintaining proper nutritional status is particularly important when the immune system is fighting disease. Indeed, subjects with severe obesity (BMI ≥ 40 kg/m^2^) are a group at high risk for COVID-19 complications [138]. Obesity is the abundance of adipose tissue, which produces cytokines and triggers an inflammatory environment [139]. In addition, physiologically, obese subjects have reduced expiratory reserve, functional capacity of the lungs, and compliance of the respiratory system. In patients with more abdominal fat, the reduced range of motion of the diaphragm further reduces respiratory function in the supine position, making ventilation more difficult [140]. In addition, obesity is also one of the most important factors affecting the severity of lung disease in COVID-19, which may lead to the famous “cytokine storm” associated with acute respiratory distress syndrome and multiple organ failure [141]. In this complex scenario, the inflammatory state of obese individuals may further exacerbate the inflammation caused by COVID-19, and hence obese individuals may be exposed to higher concentrations of proinflammatory cytokines than individuals with a normal weight [141]. Furthermore, it is important to follow a healthy diet, as gene expression levels of all cytokines are increased by excess food [142] and may modulate the processes of inflammation and oxidative stress [142].

It is interesting to note that the number of people who smoked more than 10 cigarettes a day decreased by 0.5% during this pandemic. This phenomenon can be explained by the fear mounting among smokers who are more likely to develop respiratory distress and death when they are infected with COVID-19 [143]. During quarantine, it was expected that the consumption of fresh foods would decrease, resulting in deficiencies in vitamins and minerals, including vitamins C and E and beta-carotene, which have antioxidant and anti-inflammatory properties. Deficiencies in these micronutrients have been associated with both obesity and impaired immune responses, leading to higher susceptibility to viral infections [144,145]. However, during the lockdown in Italy, people took more care to eat a MD, which is highly nutritious, particularly in northern and central Italy, which are regions in which people have a lower average BMI than people in southern Italy and the islands [146]. Obesity is a state of chronic mild inflammation that is associated with adipokine secretion in adipose tissue which causes immunomodulatory effects [145] and contributes to the development of several metabolic diseases (including insulin resistance, type 2 diabetes, and dyslipidemia). Obesity makes the immune system more vulnerable to infection by downregulating innate and adaptive immune responses, resulting in patients being less responsive to vaccinations, antivirals, and antimicrobials [145]. These immunomodulatory effects may contribute to the exacerbation of respiratory viral infections [141].

Therefore, reduction in the consumption of junk food is highly recommended in order to reduce the “obese environment” that predisposes people to weight gain and susceptibility to COVID-19 [147,148]. A diet rich in nutrients with antioxidant and anti-inflammatory properties, such as a MD, has been shown to help reduce SARS-CoV-2 toxicity [149,150].

## 17. Conclusions

Currently, we do not have sufficient evidence of supplement usage for or against COVID-19. Therefore, we cannot recommend those beyond the correction of the deficiencies. However, there are many studies suggesting a potential beneficial effect of supplements and diet from basic research to clinical studies. Larger RCT studies should be conducted to determine the significance in the COVID-19 individually.

Diet and lifestyle modification should be important against COVID-19 because of the many lifestyle-related illnesses such as obesity, physical inactivity, hypertension, and diabetes [23,24], even though we do not have direct evidence.

In addition, a rapid increase of cases with persistent symptoms after recovery from COVID-19 (Long COVID) have been gaining attention recently, which may be related with micronutrient deficiency. Supplements may be the appropriate treatment option for them, although this effect should be validated by clinical studies.

Finally, it is thought that unhealthy eating habits resulting from self-restraint, lockdowns, and quarantines during the pandemic may increase the risk of noncommunicable diseases, and this should also be kept in mind. We have accumulating evidence that COVID-19 patients are more likely to develop diabetes [151]. Nutrition therapy is recommended as part of the care for the patients even after they have recovered from COVID-19.

Correction of the nutritional deficiencies are important, regardless of infection with COVID-19 or not. Therefore, constantly checking for nutritional deficiencies and providing optimal nutritional supplementation is an important step in the optimal functioning of one’s body, including the immune system.

As malnourished patients are more likely to come from less advantaged socioeconomic groups, nutritional support is important not only for at-risk groups, but also for older individuals with relatively weak immune systems, as well as for economically challenged groups. As a community, it is important to promote not only vaccination, but also dietary support. especially for economically challenged groups against not only COVID-19, but also other health issues that will emerge in the future.

## Data Availability

The collected data files are available on request from the corresponding author, S.H.: hibino@cgt.med.osaka-u.ac.jp.

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
