# Peer review of "Modifiable Host Factors for the Prevention and Treatment of COVID-19: Diet and Lifestyle/Diet and Lifestyle Factors in the Prevention of COVID-19"

_nutrients, 2022, doi:10.3390/nu14091876_

Round 1
Reviewer 1 Report
The present study describes the main nutrients found in the literature that could help treat COVID-19.
The importance of this type of study is that it helps us to prevent or treat future viral infections or even variations of COVID-19 itself.
However, the study has little criticism and to date, there is a lot of information regarding this topic, including more than 300 reviews on the Pubmed platform.
The authors should also consider the following observations:
It would be better if the corresponding author provided an institutional mail.
Check spaces between words e.g. lines 12, 33, 74, 75, 163.
Why is a sentence highlighted in color? Line 53 as well as some words in bold, please correct if not necessary.
Move the title to the next page, line 146.
The authors describe the Mediterranean diet as a dietary pattern that would help patients with COVID-19. However Mediterranean countries such as Spain, Italy, and Greece had a high number of infections, they could explain this in more detail and critically.
Correct score from et al e.g. lines 189.
The analysis made between rice consumption and COVID-19 infected patients is interesting, however in the 15 countries analyzed, there are still countries that consume rice daily and yet maintain a high number of infections. In addition, Asian countries that had a low number of infections, such as China, was because they maintained strong containment measures. Please explain in detail this relationship or what would be the bioactive ingredient in rice that could be associated with low COVID-19 infection.
Correct the references that are written after the period, for example, line 208.
COVID-19 articles are published exponentially, including diet studies. Interestingly, there are no studies with negative results in this review, with the exception of omega-3, where the consumption doses and effect of this compound are mentioned. Both studies that have worked and those that have not reported a change should always be cited to give a further analysis of the study.
Valeriana officinalis should be written in italics, line 471.
Author Response
The authors should also consider the following observations:
It would be better if the corresponding author provided an institutional mail.
I changed my E-mail as an institutional mail to hibino@cgt.med.osaka-u.ac.jp.
Check spaces between words e.g. lines 12, 33, 74, 75, 163.
I corrected the space the line that was pointed out. I corrected these minor points.The authors describe the Mediterranean diet as a dietary pattern that would help patients with COVID-19. However Mediterranean countries such as Spain, Italy, and Greece had a high number of infections, they could explain this in more detail and critically.
The analysis made between rice consumption and COVID-19 infected patients is interesting, however in the 15 countries analyzed, there are still countries that consume rice daily and yet maintain a high number of infections. In addition, Asian countries that had a low number of infections, such as China, was because they maintained strong containment measures. Please explain in detail this relationship or what would be the bioactive ingredient in rice that could be associated with low COVID-19 infection.COVID-19 articles are published exponentially, including diet studies. Interestingly, there are no studies with negative results in this review, with the exception of omega-3, where the consumption doses and effect of this compound are mentioned. Both studies that have worked and those that have not reported a change should always be cited to give a further analysis of the study.
I added positive and negative information. Furthermore, it was repeatedly made clear that there was no clear evidence at this time. I rewrote the whole sentence, clearly explaining that there was no evidence. Since it has changed significantly, it is written in red in the text, so please review it.
Reviewer 2 Report
With great interest, I read the manuscript on Modifiable Host Factors for the Prevention and Treatment of COVID-19 regarding Diet and Lifestyle factors in the prevention of COVID-19. The authors presented many lifestyles and nutritional factors that have been linked to the recent COVID-19 epidemic. In presenting the results obtained by reviewing the scientific literature, some parts are nicely presented, but some parts of the manuscript require improvement. For example, I suggest that the introductory part related to the introduction of SARS-CoV-2 be shortened and that the possible connection with lifestyle and nutritional factors, especially functional ones, be addressed more. It may not take as much detail about SARS-CoV-2 types, because the virus is currently still going through its mutations. I also suggest making a table that would visually show all the presented lifestyle and nutritional factors assessed by the strength of the association with COVID-19 (negative and positive association) and give a brief remark whether it is a review article or cross-sectional studies on a specific subject. Also, the conclusion should be more related to nutritional factors, ie more to functional food ingredients, as mentioned in the summary. More specifically, in some places the authors did not provide references, so I advise you to review all the above statements in detail once again. For example, line 64 (Preliminary clinical studies ...), line 103-105 and line 15-108, line 131-133, line 143-145 (the sentence sounds like the authors' opinion, but it could be another one's conclusion, it is confusing), line 171-173 (the sentence needs a reference / s for the statement because there is no universal definition of an anti-inflammatory diet yet, the authors could elaborate this issue and connect this with the implications to COVID-19 infection and consequent cytokine storm alleviation with food components that have potential anti-inflammatory effects. In conclusion, the authors should carefully review and edit their statements and other ones' conclusions presented in their review.
Author Response
I have corrected the part you pointed out.
And I added positive and negative information. Furthermore, it was repeatedly made clear that there was no clear evidence at this time.  Since it has changed significantly, it is written in red in the text, so please review it.

Reviewer 3 Report
Proposed paper is well written and could be interesting. However it is seriously limited by the fact that fundamentally there is no prove of the benefit on supplements in the care of COVID-19 infection. Although we are now in another phase of the epidemia compared with march 2020 authors have to be very cautious when speculating on the fact that some nutritional supplements could save life in SARS-CoV-2 related disease. Many important drugs have been tried (among other hydroxychloroquine) and no results have been found in well designed trial.
Most of the reference used are speculative on potential benefit derived from nutritional supplements. For vitamin C a case-report and a case series of 17 cases is used to advice on its supplementation.
Furthermore although some deficiencies (selenium, vitamin D) have been clearly related to worst outcomes no evidence that their supplementation could determine a better outcome exist.
In my opinion paper should be strongly downgraded to a description of what is surely now on the topic (deficiencies and worst outcomes) and the few studies that showed benefit of some supplementation but clearly defining their limitation (few patients). Finally, some suggestion (that have to be only suggestion) can be drawn also based on what is known on other viruses infection.
Although, nutritional medicine is something fundamental for well-being, I think in this phase it could be dangerous to let people think that we can cure COVID with vitamin C or D without any evidence of this. Since all the pubblication on COVID are used by non scientific communicators I confirm to down-grade the paper as previously suggested.
Furthermore section on alchool disinfectant, mask, vaccines and current dominant variants are not related to the topic and should be deleted.
Author Response
Reviewer 3
Proposed paper is well written and could be interesting. However it is seriously limited by the fact that fundamentally there is no prove of the benefit on supplements in the care of COVID-19 infection. Although we are now in another phase of the epidemia compared with march 2020 authors have to be very cautious when speculating on the fact that some nutritional supplements could save life in SARS-CoV-2 related disease. Many important drugs have been tried (among other hydroxychloroquine) and no results have been found in well-designed trial.Most of the reference used are speculative on potential benefit derived from nutritional supplements. For vitamin C a case-report and a case series of 17 cases is used to advice on its supplementation.Furthermore although some deficiencies (selenium, vitamin D) have been clearly related to worst outcomes no evidence that their supplementation could determine a better outcome exist.In my opinion paper should be strongly downgraded to a description of what is surely now on the topic (deficiencies and worst outcomes) and the few studies that showed benefit of some supplementation but clearly defining their limitation (few patients). Finally, some suggestion (that have to be only suggestion) can be drawn also based on what is known on other viruses infection.Although, nutritional medicine is something fundamental for well-being, I think in this phase it could be dangerous to let people think that we can cure COVID with vitamin C or D without any evidence of this. Since all the pubblication on COVID are used by nonscientific communicators I confirm to down-grade the paper as previously suggested.
As you pointed out, I tried not to write too positively about unfounded treatments. I also strongly emphasized the lack of evidence.
Furthermore section on alchool disinfectant, mask, vaccines and current dominant variants are not related to the topic and should be deleted.
I deleted vaccines and current dominant variants.
I added positive and negative information. Furthermore, it was repeatedly made clear that there was no clear evidence at this time. I rewrote the whole sentence, clearly explaining that there was no evidence. Since it has changed significantly, it is written in red in the text, so please review it.
Thank you very much for your consideration in this matter.
Best regards,
Sawako Hibino

Round 2
Reviewer 3 Report
Authors replies to the query raised and paper improves. However, there is one last point: authors says that they have removed part on disinfection, mask, vaccines and variants but they are still in the paper. Please remove it.
Author Response
Thank you very much for reviewing.
I just have removed part on disinfection, mask, vaccines and variants in my paper. And I will ask the publisher to check my English grammar as you point out.
Please see my resubmitted manuscript.
Thank you very much for your consideration in this matter.
Best regards,
Sawako Hibino